# Baseband control of single-electron silicon spin qubits in two dimensions

Florian K. Unseld [1,3], Brennan Undseth [1,3], Eline Raymenants [1], Yuta Matsumoto[1], Sander L. de Snoo [1], Saurabh Karwal[2], Oriol Pietx-Casas[1], Alexander S. Ivlev [1], Marcel Meyer [1], Amir Sammak[2], Menno Veldhorst [1], Giordano Scappucci [1] & Lieven M. K. Vandersypen [1]✉

Micromagnet-enabled electric-dipole spin resonance (EDSR) is an established method for high-fidelity single-spin control in silicon, although so far experiments have been restricted to one-dimensional arrays. In contrast, qubit control based on hopping spins has recently emerged as a compelling alternative, with high-fidelity baseband control realized in sparse two-dimensional hole arrays in germanium. In this work, we commission a $^{28}$Si/SiGe $2 \times 2$ quantum dot array both as a four-qubit device using EDSR and as a two-qubit device using baseband hopping control. We establish a lower bound on the fidelity of the hopping gate of 99.50(6)%, which is similar to the average fidelity of the resonant gate. The hopping gate also circumvents the transient pulse-induced resonance shift from heating observed during EDSR operation. To motivate hopping spins as an attractive means of scaling silicon spin-qubit arrays, we propose an extensible nanomagnet design that enables engineered baseband control of large spin arrays.

Industrial fabrication compatibility is a flagship argument for semiconductor spin qubits in gate-defined quantum dots as a candidate for large-scale quantum computation and simulation[1–5], but this promise can only be fully realized if the spin physics utilized to control qubits is also extensible. Currently, on-chip micromagnets have enabled state-of-the-art devices to exhibit powerful primitives including high-fidelity universal gate sets[6–8], high-fidelity initialization and readout of multiple qubits[9,10], and coherent spin shuttling[11,12].

The micromagnet typically serves two purposes. It produces a longitudinal gradient parallel to the quantization axes to give each spin a unique frequency and a transversal gradient that allows microwave electric fields to drive single-qubit gates via electric-dipole spin resonance (EDSR)[13–15]. While this method of operation has been extended to 12 qubits[16], the linear charge-noise sweet spot of present magnet designs has motivated scaling in only one dimension. The limited connectivity and stringent fault tolerance thresholds of linear arrays imply that entering the second dimension is all but essential[17].

Although proposals exist for scaling EDSR control into two-dimensional arrays using on-chip magnets[18,19], an experimental demonstration beyond 1D has, until now, been lacking.

Even in 1D arrays the dissipation of microwaves used for EDSR control is known to produce heating effects that complicate multi-qubit operation[20–23]. Typically, this manifests as qubit frequency shifts that are contextual upon the magnitude and duration of preceding microwave bursts. Circumventing this issue by working at warmer device temperatures is possible, but microwaves may be bypassed altogether. Whereas singlet-triplet and exchange-only encoded qubits allow for universal baseband control, aspects like state leakage and gate complexity have limited error rates to higher levels than for the single-spin encoding. Recently, single-hole spins in a 2D germanium array have been manipulated with high fidelity using baseband hopping control exploiting large differences in the quantization axis arising from dot-to-dot variations in the $g$-tensor[24,25]. It is therefore intriguing whether such control can be engineered with on-chip

[1]QuTech and Kavli Institute of Nanoscience, Delft University of Technology, Lorentzweg 1, 2628 CJ Delft Delft, The Netherlands. [2]QuTech and Netherlands Organization for Applied Scientific Research (TNO), Stieltjesweg 1, 2628 CK Delft Delft, Netherlands. [3]These authors contributed equally: Florian K. Unseld, Brennan Undseth. ✉e-mail: L.M.K.Vandersypen@tudelft.nl

magnets and, if so, what implications this poses for future architectures.

Here, we commission a 2 × 2 silicon quantum dot device for qubit control in two distinct regimes. First, we demonstrate that conventional micromagnet-based EDSR control of all four spins is possible in the 2D array, and we also demonstrate nearest-neighbor exchange control. We benchmark the performance of the four-qubit system with a particular emphasis on the crosstalk caused by off-resonant driving. Next, we lower the external magnetic field of the half-filled array to demonstrate qubit operation via hopping gates, whereby the tip in quantization axis is induced by the engineered magnetic stray field. We further analyze the qubit coherence, fidelity, and crosstalk properties of the hopping gate. Finally, we propose how a repeated nanomagnet pattern can be used to engineer hopping control across an arbitrarily large 2D array, thereby illustrating how baseband control of single-electron spins in silicon may be a compelling control method for future devices.

## Results

### A 2 × 2 silicon quantum processor based on EDSR

The most straightforward progression from a one-dimensional silicon spin-qubit array to a two-dimensional array is the adoption of existing control strategies with only minor accommodations[9]. A 2 × 2 quantum dot array as shown in Fig. 1a was fabricated on a $^{28}$Si/SiGe heterostructure (residual nuclear spins of 800 ppm) and is used to accumulate four single electrons under each plunger gate as has been previously demonstrated[26]. After magnetizing the micromagnet at a field of +2 T along the positive y-axis as indicated in the SEM and schematic of Fig. 1a, the external field is reduced to 200 mT. We use Pauli-spin blockade (PSB) for initialization and measurement on the horizontal pairs (Q$_1$Q$_4$ and Q$_2$Q$_3$) along with electric-dipole spin resonance (EDSR) to perform addressable single-qubit rotations. The 22.5 deg rotation of the 2 × 2 array combined with the longitudinal field gradient gives good spectral separation of the spins with a minimum frequency difference of 74 MHz. Using adequate parameters we find reasonable agreement with micromagnet simulations (see Supplementary Fig. 1 for details). The Rabi frequency of all four spins was tuned to 2 MHz to stay within the linear amplitude scaling regime (see Supplementary Fig. 8) to observe the Chevron patterns in Fig. 1b.

Despite the qubits not lying along the decoherence sweet axis of the micromagnet, we observe coherence metrics similar to[9]. The measured Ramsey decay times are $T_2^* = \{3.31(9)\,\mu s, 2.03(2)\,\mu s, 3.57(8)\,\mu s, 2.90(5)\,\mu s\}$ for qubits Q$_1$ through Q$_4$ respectively with integration times of around 30 min. Similarly, we measured Hahn-echo decay times of $T_2^H = \{30.21(24)\,\mu s, 16.22(18)\,\mu s, 40.38(21)\,\mu s, 21.77(38)\,\mu s\}$. Notably, the trend of coherence times is not strictly correlated to the designed decoherence gradients at each dot location (see Supplementary Fig. 1). There are three possible reasons for such discrepancies. First, the decoherence gradient is not isotropic in the plane of the quantum well, and the observed coherence times will depend on the microscopic orientation of the local charge noise fluctuators for each qubit. Second, large variations in coherence may occur for sufficiently sparse charge noise baths depending on the nature of the constituent fluctuators[27]. Third, hyperfine noise due to residual nuclear spins in the heterostructure may also contribute to decoherence that does not couple to the spins via the micromagnet gradient[28].

Two-qubit interactions are controlled using the barrier gates located between neighboring plunger gate pairs to modulate the Ising-

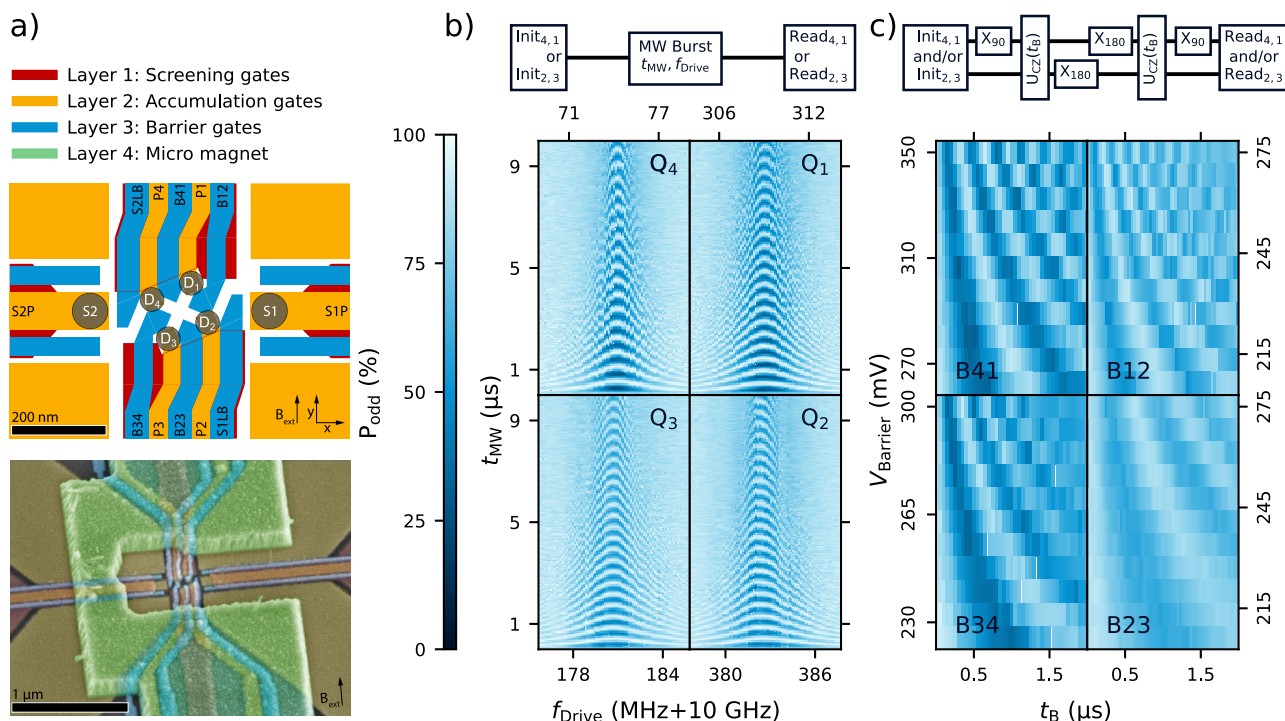

**Fig. 1 | A 2 × 2 silicon spin qubit array. a** Design schematic and false colored SEM image of a device nominally identical to the one measured. In the schematic, single-electron transistor (SET) sensor positions are marked with S1 and S2, and the quantum dots are labeled clockwise with D$_1$ through D$_4$. We refer to each qubit as Q$_1$ through Q$_4$ according to the dot it primarily inhabits. The SEM image includes the micro magnet on top of the gate stack (not shown in the schematic). **b** Odd parity probability after driving the qubit with a microwave burst as indicated by the block diagram. The respective Chevron patterns of all qubits are arranged according to their physical position. The drive power and modulation amplitude were adjusted per qubit to achieve a Rabi frequency around 2 MHz. **c** Decoupled controlled phase oscillations of adjacent qubit pairs using the pulse sequence indicated in the block diagram. The Ising-like interaction $U_{CZ}$ is controlled by the voltage $V_{Barrier}$ on the barrier gate located between the respective plunger gates. We chose a static operating point such that the extrapolated exchange strength $J_{off}$ <20 kHz and apply a pulse on the barriers large enough such that $J_{on}$ >1 MHz. The large amplitudes for these barrier pulses as well as the particular fanout of gates B34, P3, and B23 often caused substantial degradation of the readout signal (see "Discussion").

like exchange interaction. We observe the characteristic CPhase oscillations for all four nearest-neighbor pairs of qubits as a function of time and barrier pulse height as depicted in Fig. 1c. Although we operate at the symmetry point to minimize the sensitivity to charge noise, the quality factor $T_2^{\text{DCZ}}/t_{\text{CZ}}$ of the oscillations for all pairs is limited to $Q_{2Q} = \{17(2), 7.5(7), 7.1(4), 12.2(7)\}$ for B12, B23, B34, and B41 respectively, which bounds the achievable two-qubit gate fidelities to modest values. Supplementary Fig. 3 shows state tomography results before and after applying a calibrated CZ gate for qubit pairs $Q_4Q_1$ and $Q_2Q_3$, highlighting the universal capability of this processor.

One reason for the suboptimal two-qubit performance may be the magnitude of pulses, ranging from 275 mV to 350 mV, required to achieve a satisfactory on/off exchange ratio as a result of the small barrier lever arms. Such large pulses jeopardize device stability and, in some cases, degrade readout visibility (see Fig. 1c). Although we apply large pulse amplitudes, the maximum achievable exchange couplings of 1 MHz-4 MHz are small with respect to the dephasing times of the qubits.

Despite this shortcoming, we can still draw valuable insights about multi-qubit control with such an EDSR-based architecture. The quality factors $T_2^{\text{Rabi}}/t_{180}$ of the four qubits are measured to be $Q_{1Q} = \{64(4), 67(6), 65(5), 64(7)\}$, and we estimate the resonant $X_{90}$ fidelity averaged across all four qubits to be 99.54(4)% using randomized benchmarking (RB) as displayed in Fig. 2a. By comparing the infidelity expected from both quasi-static qubit frequency fluctuations ($T_2^*$) and decoherence of the spins subject to driving ($T_2^{\text{Rabi}}$), we conclude that it is the latter that predominantly limits the resonant single-qubit gate fidelity we observe from randomized benchmarking (see Supplementary Note 5 for details). It is possible that higher fidelities could be achieved by taking advantage of the higher possible Rabi frequencies on some of the qubits (see Supplementary Fig. 8). Having established a good single-qubit gate set for each qubit, we can probe

the crosstalk such gates impart on neighboring qubits. Controlling crosstalk is critical for multi-qubit operation, and while the absolute amount of crosstalk is relevant for calibration, its contextuality in time and number of gates is particularly indicative of how difficult this calibration becomes in practice. For EDSR, crosstalk commonly manifests as a spurious pulse-induced resonance shift (PIRS). We probe this "heating effect" on idling qubits using a modified Hahn echo sequence as depicted in Fig. 2b[21]. The sequence includes an off-resonant microwave burst that simulates the drive of another qubit, and we quantify the magnitude of this burst in relation to the average power required to perform single-qubit rotations across all four qubits in this device.

A nontrivial transient phase pickup is observed for all qubits for off-resonant bursts that are energetically comparable to those used for single-qubit gates. A particular instance is shown in Fig. 2c, though all qubits exhibit the same qualitative behavior (see Supplementary Fig. 9). With increasing delay time and amplitude, more phase is picked up until an apparent saturation is reached. This transient response has been documented in previous studies[21]. However, when the off-resonant burst occurs closer in time to the resonant echo pulse, less phase is picked up. The most likely origin of this counterintuitive behavior is the transient heating caused by the resonant echo pulse: the heating induced by the decoupling pulse reduces the transient effect of the off-resonant burst. The nonlinear combination of the two transients has severe consequences for crosstalk calibration: the required compensation will be sensitive to both pulse scheduling and amplitude.

It has previously been observed that PIRS is dependent on the temperature of the device[22]. To verify this, we repeat the above experiment on all qubits for different mixing chamber temperatures. From the fitted data (e.g. Fig. 2c) we extract and compare the maximum phase pickup as a function of temperature. Figure 2d shows how

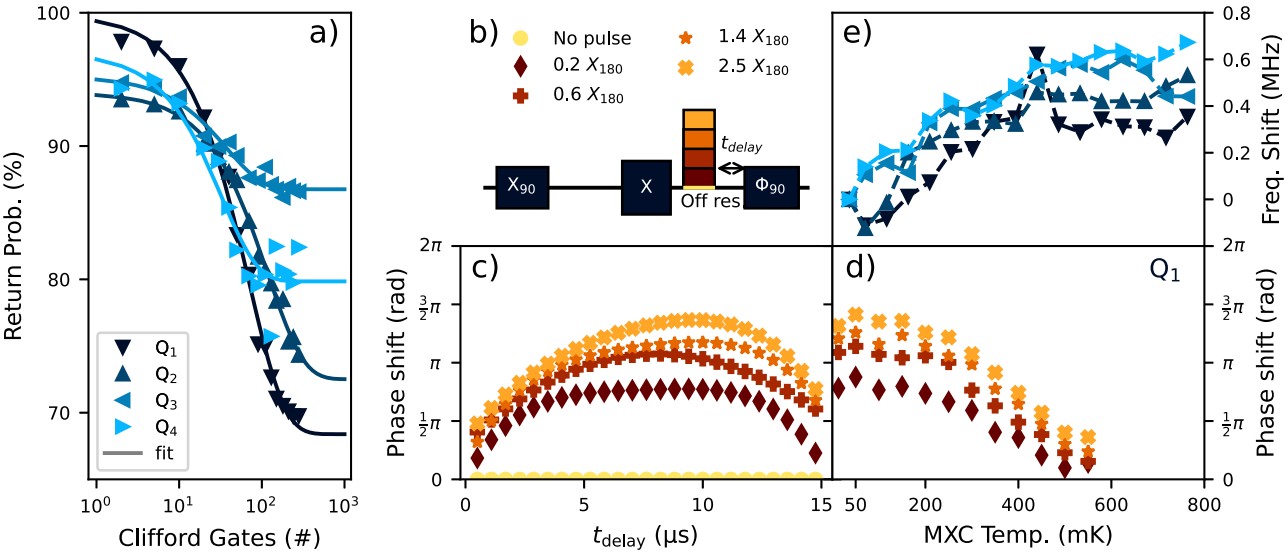

**Fig. 2 | EDSR Operation and PIRS Effects. a** Return probability as function of the number of applied Clifford gates averaged over twenty different gate sequences for all four qubits. As each qubit readout has a different visibility, the curves decay to different values. The fitted Clifford gate fidelities are $F_{\text{Clif}}^{\text{res}} = \{99.28(5)\%, 99.58(4)\%, 98.7(2)\%, 98.5(3)\%\}$ for $Q_1$ through $Q_4$ respectively. The Clifford gate set is compiled using only $X_{\pm 90}$ and $Y_{\pm 90}$ primitive gates (see Methods), and the corresponding average physical gate fidelities are, respectively, $F_{\text{avg}}^{\text{res}} = \{99.67(2)\%, 99.80(2)\%, 99.39(8)\%, 99.31(15)\%\}$. **b** Adapted Hahn Echo sequence to measure crosstalk effects as introduced in ref. 21. The off-resonant burst simulates the drive of a different qubit. By substituting the last $X_{90}$ with a $\Phi_{90} = X_{90}Z(\Phi)$ operation in the echo sequence and sweeping the phase $\Phi$, the relative phase pickup due to the off-resonant burst can be inferred. The transient

behavior is detected by varying the temporal position $t_{\text{delay}}$ of the off-resonant burst with respect to $\Phi_{90}$. **c** Example of phase pickup as a function of $t_{\text{delay}}$ for $Q_1$ at 100 mK for different amplitudes of the off-resonant burst ((**b**) for legend). We use a standard Hahn-echo sequence (yellow data points) as a reference to remove constant artifacts introduced by the echo pulse sequence. **d** The maximum phase shift extracted from measurements as in (**c**) as a function of mixing chamber temperature for qubit $Q_1$ for four different off-resonance bursts ((**b**) for legend). All qubits exhibit similar behavior (see Supplementary Fig. 9). **e** Temperature dependence of the bare qubit frequencies as measured by a Ramsey experiment relative to the frequency at base temperature. Symbols refer to different qubits as in (**a**).

increasing the mixing chamber temperature reduces the magnitude of the PIRS effect for $Q_1$ (see Supplementary Fig. 9 for $Q_2$-$Q_4$). This substantial temperature increase, however, also has an impact on qubit coherence. Although the $T_2^*$ of all qubits is relatively unaffected by temperature, the $T_2^H$ generally decreases monotonically as the mixing chamber temperature is raised (see Supplementary Fig. 11). Additionally, an increase in spin Larmor frequency as a function of temperature is observed for all quantum dots as seen in Fig. 2e.

While these results are qualitatively in agreement with a previous study of a six-spin array[22], there are three notable differences. First, the generally monotonic increase in frequency of all four qubits studied here contrasts with the previous measurement of a non-monotonic trend across the six-qubit array. Second, the heating effect is more moderate than the 1 MHz frequency shifts observed by Undseth et al. when using similar microwave bursts[22]. In this work, PIRS was observed through residual phase pick-up corresponding to a frequency shift on the order of 10–100 kHz. Last, the measured PIRS phase pickup is consistent with a negative Larmor frequency shift opposite to the frequency shift seen when raising the device temperature. This suggests that although the device temperature clearly plays a meaningful role in how PIRS manifests, the dissipated heat of the microwave burst may not be the only PIRS mechanism in this device. It has been proposed that thermally-sensitive environmental fluctuators can significantly contribute to the temperature dependence of the qubit frequencies[29]. Our observations underscore the difficulty of predicting and compensating multi-qubit crosstalk with resonant control.

## Baseband operation

In light of the crosstalk challenges arising from microwave control, the operation of spin qubits using exclusively baseband pulses makes for a compelling alternative while retaining the Loss-DiVincenzo qubit encoding. Such control was recently demonstrated by hopping a hole spin qubit between germanium quantum dots with different quantization axes[25] and bears resemblance to the strong-driving flopping mode regime proposed by ref. 30. Similarly, the 2 × 2 silicon array allows us to take advantage of quantization axis variation originating from the inhomogeneous magnetic field introduced by the micromagnet. However, even in the 2 × 2 array a large external field forces the spins' quantization axes to be nearly colinear. In order to accentuate the inhomogeneous magnetic field and limit Larmor frequencies for baseband control, the external field is reduced.

First, the micromagnet is magnetized in a 0.8 T external field. We populate the array with a single electron in dots 1 and 4, allowing us to keep PSB readout on the pair $Q_4 Q_1$ while dots $D_2$ and $D_3$ remain empty. We track the qubit frequencies of $Q_1$ and $Q_4$ using adiabatic inversion pulses as the external field is reduced step-by-step as shown in Fig. 3a. We observe the qubit frequencies dropping both as a result of the lower external field as well as the demagnetization of the micromagnet. The polarization of the micromagnet is inferred by fitting the measured frequencies to a simplified magnet model where the magnetization is assumed to be homogeneous with no crystalline domains or shape anisotropy coming into consideration (see Supplementary Note 2). Even so, this model provides a firm basis for understanding the qualitative qubit properties across a wide range of external field settings.

Even at zero external field, the remanence of the micromagnet forces the quantization axes to be reasonably colinear and provides a driving gradient large enough for conventional EDSR operation. Notably, the Hahn-echo coherence times of $Q_1$ and $Q_4$ nearly double from 30.2 μs and 21.3 μs to 49.1 μs and 43.8 μs respectively as shown in Fig. 3b. Curiously, the $T_2^*$ coherence times do not change significantly with the magnetic field and decoherence gradient. This suggests the limiting low-frequency noise may be different than the noise inducing Hahn-echo decay. The amplitude with which these spins are driven with $f_{Rabi} \approx 2$ MHz also increases by about a factor of 3. These

observations are compatible with the estimated change in decoherence and driving gradients at the dot locations: Fig. 3c shows that the decoherence gradient permitting charge noise to couple to the spins has approximately halved while Fig. 3d shows the driving gradient has been reduced by a factor of 3. As predicted by simulations, the 100 MHz frequency separation between $Q_1$ and $Q_4$ is retained and becomes the main contribution to the decoherence gradient. This suggests that EDSR operation in the absence of an external solenoid, a mundane but material obstacle for the near-term scalability of semiconductor spin qubits, is a realistic possibility.

Decreasing the external field further, we reach a regime where the local magnetic field in the dot region is increasingly inhomogeneous owing to the competition between the remaining micromagnet polarization and the opposed external field. The spin quantisation axes at the quantum dots will no longer align but instead point in different directions. This is most clearly illustrated by rendering the quantization axes at each dot location as shown in Fig. 3e). Despite the extreme inhomogeneity and low effective polarization of the micromagnet, we observe no pronounced instability of the spin frequencies. On the contrary, the measured Ramsey and Hahn-echo decay times universally improve upon the coherence measured at $B_{ext}$ = 200 mT with EDSR control (Fig. 3b). However, at a nominal external field setting of $B_{ext}$ = −15.8 mT the driving gradient is too small for resonant control. Fortunately, the large quantization axis tips enable coherent control by hopping the electrons from dots $D_1$ and $D_4$ to the vacant dots $D_2$ and $D_3$. Furthermore, the $Q_1$ and $Q_4$ Larmor frequencies of 131 MHz and 103 MHz respectively are practical for arbitrary waveform generator (AWG) control.

Figure 4a illustrates the pulse sequence required for a hopping gate between dots $D_1$ and $D_2$. During each ramp, the charge transfer between dots should be adiabatic (see Fig. 4b), hence a sufficiently large tunnel coupling $t_c$ should be maintained between dots. A fast detuning ramp $v = \epsilon / t_{ramp}$ is desirable to minimize dephasing while the spin is most strongly hybridized with the charge at $\epsilon = 0$ μeV. Within the subspace of the adiabatic charge transition, the spin state transfer should be diabatic (see Fig. 4c) which is ensured by traversing the Hamiltonian "step" sufficiently fast. Provided these conditions are met (see Supplementary Note 6), a universal gate set may be calibrated with the appropriate choice of parameters $t_1$, $t_2$, and $t_{add}$[25]. One repetition of two cycles as drawn is sufficient to calibrate an $X_{90}$ provided the quantization axis tip between dots lies between 22.5 deg and 157.5 deg. For quantization axis tips between 45 deg and 135 deg, one repetition of a single cycle suffices. The additional idling time parameter $t_{add}$ between subsequent repetitions ensures the axis of rotation is consistent when concatenating multiple $X_{90}$ gates.

To quantify the quantization axis tip experimentally, we apply multiple repetitions of spin hopping for varying times $t_1$ and $t_2$ and measure the resulting spin fraction. For the presented data the ramp time between $t_1$ and $t_2$ was set to be 2 ns. This is comparable to the minimum rise time of the AWG output signal and was chosen to minimize decoherence during the shuttling process. The tunnel coupling was estimated to be roughly 48 μeV for all hopping gate experiments unless otherwise noted (see Supplementary Fig. 12). The measured pattern sensitively depends on the tip angle and may be fit numerically to the expected unitary evolution of the spin (see Methods). An example of a 2-cycle-4-repetition-protocol measurement and fit is shown in Fig. 4d and yields a tip angle of $\theta_{1,2} = 37.3(2)$ deg. Similarly, we measure $\theta_{4,3} = 47.5(2)$ deg (see Supplementary Fig. 4). In the latter case, we observe that pulsing on P3 causes severe degradation to the readout signal (as was also the case for probing two-qubit interactions in Fig. 1). We therefore opt to only benchmark the $Q_1$ hopping gate.

To coarsely calibrate the $t_1$, $t_2$ and $t_{add}$ idling times for a high-fidelity single-qubit gate, we select parameters where the $X_{90}$ fidelity is theoretically maximized as shown in Fig. 4d. We then fine tune the

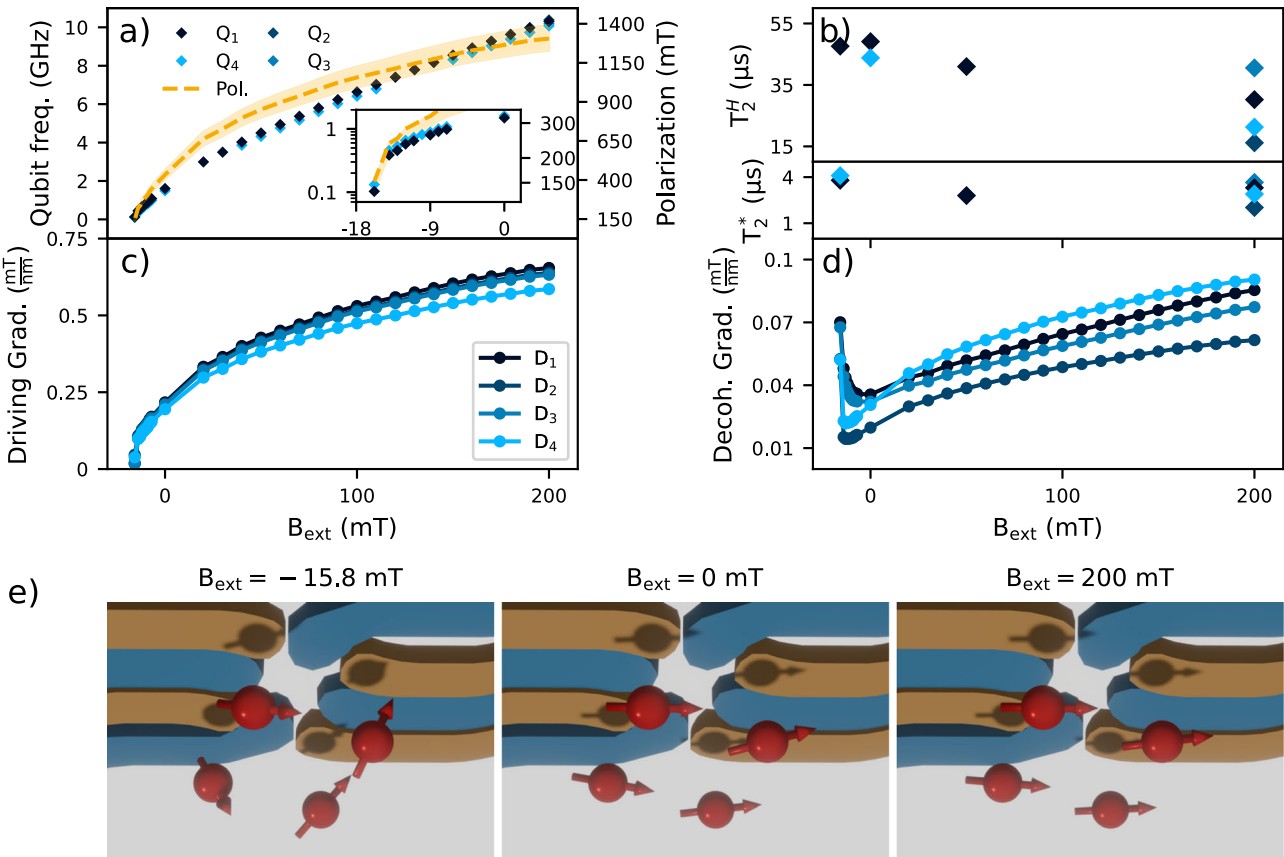

**Fig. 3 | Magnetic field dependence of device and qubit properties.**
**a** Dependence of the qubit frequencies on the external magnetic field. Adiabatic inversion pulses are used for external fields above $B_{ext} = -15$ mT (qubit frequencies above 400 MHz) while Ramsey experiments with hopping control is used for lower fields. The dashed line shows the predicted homogeneous polarization of the micromagnet as extracted by fitting the measured qubit frequencies to a simplified micromagnet model (see Supplementary Note 2). The shaded area provides a generous estimate of uncertainty by displacing the micromagnet model $\pm 15$ nm out-of-plane and repeating the fits. The inset zooms in onto the behavior below $B_{ext} = 0$ mT with the qubit frequencies and polarization plotted on a logarithmic scale. **b** (lower) $T_2^*$ integrated for 30 min at various external field settings. Ramsey decay times are relatively unchanged by the magnitude of the external field and, consequently, the decoherence gradient. Symbols as in (**a**). **b** (upper) $T_2^H$ integrated for 30 min at various external field settings showing an inverse relation to external field and decoherence gradient. From the extracted micromagnet

polarization in (**a**) we extract (**c**) the driving gradient along the $y$-axis and (**d**) the decoherence gradient magnitude at the center of the plunger gates. The driving gradient decreases monotonically while the decoherence gradient shows a sweet spot close to $B_{ext} = 0$ mT. **e** Renders of the plunger and barrier gates (yellow, blue) from below with the four electron spins (red, from left-to-right: $Q_3$, $Q_2$, $Q_4$, $Q_1$) for three different external field settings. The arrows indicate the quantization axes at the dot location which, approximating an isotropic $g$-factor of 2 in the silicon quantum well, is effectively assumed to be colinear with the total magnetic field vector. At $B_{ext} = 200$ mT the quantization axes are almost perfectly aligned with each other within a few degrees. At $B_{ext} = 0$ mT, some deviation becomes evident. At $B_{ext} = -15.8$ mT the differences in quantization axes are very pronounced. Based on the micromagnet simulation, the tips are estimated to be between about 30 deg and 60 deg depending on the exact locations of the quantum dot in the micromagnet stray field.

parameters by running a sequence of $X_{90}$ gates and select the precession times that produce a periodicity of 4 as exemplified for $t_1$ in Fig. 4e. Consequently, we observe discretized Rabi oscillations of around 5 MHz. By padding idling times with $2\pi$ rotations, we intentionally do not operate the gates as fast as is theoretically possible in order to minimize artifacts that occur at the limit of the AWG time resolution. Z-rotations are implemented with a physical wait time calibrated by a Ramsey measurement of the spin Larmor frequency. Constructing a Clifford gate set comprised of $X_{90}$ hopping gates and physical Z-rotations, single-qubit randomized benchmarking of $Q_1$ yields an average Clifford gate fidelity of about $F_{Clif, odd}^{hop} = 99.01(11)\%$ and $F_{Clif, even}^{hop} = 99.49(7)\%$ as shown in Fig. 4f. By attributing all errors to the $X_{90}$ hopping gate we can provide a lower bound on its fidelity $F_{X_{90}}^{hop} = 99.50(6)\%$ (see Methods). This is directly comparable to the physical resonant gate fidelities achieved with EDSR control at high field. Using blind randomized benchmarking, we estimate that the hopping gate fidelity is bounded by the tunnel coupling tunability in this device leading to non-adiabatic charge shuttling and state leakage (see Supplementary Notes 6 and 7 for further discussion)[31]. When

applied to the data of Fig. 4f, we obtain a leakage rate of 0.13(5)% and a corresponding Clifford gate fidelity of $F_{Cliff}^{hop, leak} = 99.26(5)\%$.

We may also compare the performance of the EDSR and hopping mechanisms with regards to crosstalk (Fig. 4g). We employ the same PIRS methodology as in Fig. 2 and perform a Hahn echo sequence on qubit $Q_1$ using hopping gates. As $Q_4$ is used for parity readout, we calibrate an $X_{180}$ hopping gate on this qubit and embed it within the echo sequence (in place of the off-resonant burst) to measure the resulting phase pickup from the physical shuttling. In contrast to the EDSR-based gates, the hopping mechanism imparts a constant phase pick-up regardless of its temporal position in the echo sequence. We attribute this shift to the change in electrostatic conditions caused by both the gate voltage pulses and movement of the $Q_4$ electron in the quantum well which shifts the position of $Q_1$ in the magnetic field gradient. The qubit pair is maintained with very little residual exchange, hence negligible conditional phase is acquired during the sequence.

We also introduce a 10 GHz burst commensurate with an $X_{90}$ gate in the standard EDSR regime. We observe a transient phase pickup

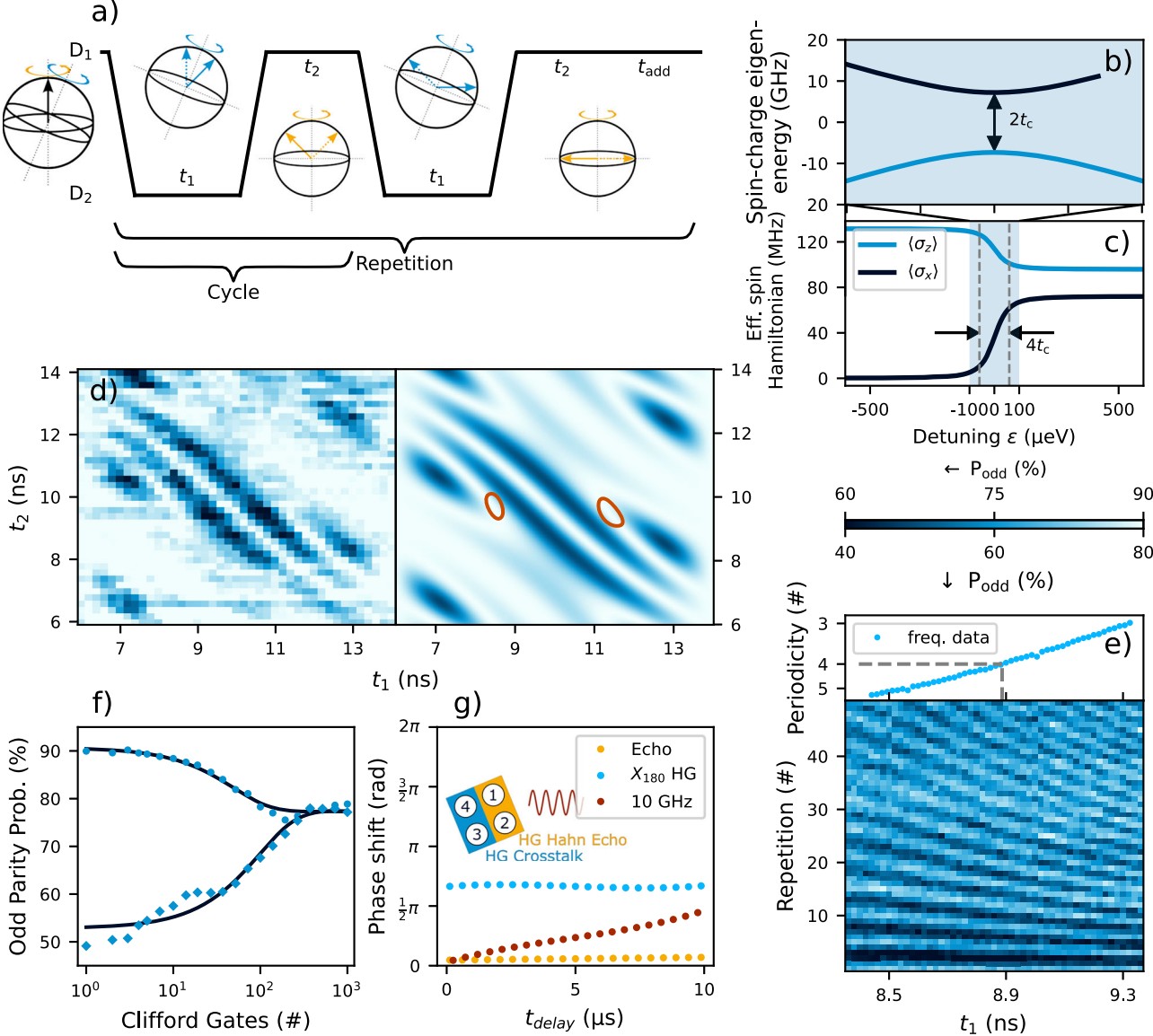

**Fig. 4 | Implementation and characterization of a Hopping Gate. a** Hopping sequence and sketched evolution of the spin state vector. The dashed (solid) arrow indicates the initial (final) vector at each step while yellow (blue) correspond to the original (secondary) quantum dot. The sequence results in a 90 deg rotation around the x-axis in $D_1$. **b** Spin-charge energy level diagram for the $D_{1,2}$ double-dot system illustrating the charge avoided-crossing. **c** Components of the effective spin qubit Hamiltonian $H\prime$ with respect to the quantization axis of $D_1$ in the subspace of the charge ground state given by $\langle \sigma_i \rangle = \mathrm{Tr}(\sigma_i H\prime)$ for the $D_{1,2}$ double-dot system (see Supplementary Note 6). **d** Experimental (left) and fitted (right) odd-parity probability using two cycles and four repetitions (refer to (**a**)) while sweeping $t_1$ and $t_2$ for the $D_{1,2}$ system with $t_{\mathrm{add}}$ fixed to 3.6 ns. The quantization axis tip is fit to 37.3(2) deg (see Methods for details). The red regions highlight the $t_1$ and $t_2$ where each one of the four repetitions achieves a high fidelity $X_{90}$ gate. The tunnel coupling for this measurement was estimated to be roughly 18 $\mu$eV. Additional data, fits and fidelity

contours for qubits $Q_1$ and $Q_4$ are shown in Supplementary Fig. 4. **e** Fine calibration of $t_1$ by repeating the roughly calibrated gate many times with minor adjustments to one of the timing parameters. We extract the periodicity of the pattern and select the $t_1$ that matches a periodicity of four for the benchmarked gate. The difference in $t_1$ with respect to the data in (**c**) is due to adjustments in $t_{\mathrm{add}}$. **f** Randomized benchmarking using both initial basis states. The Clifford set is compiled using only $X_{90}$ hopping gates and physical Z-rotations (see Methods). The fitted Clifford gate fidelities are $F_{\mathrm{Clif}}^{\mathrm{hop, odd}} = 99.01(11)\%$ and $F_{\mathrm{Clif}}^{\mathrm{hop, even}} = 99.49(7)\%$ for the odd and even parity inputs respectively (see Methods). **g** PIRS-like measurements showing the phase accumulation after a hopping gate and a 10 GHz burst when included in an echo sequence as depicted in Fig. 2b. The hopping gate is implemented as an $X_{180}$ gate on $Q_4$. The microwave burst carries the same energy as an average $X_{90}$ gate in this device during EDSR operation. Raw data is shown in Supplementary Fig. 10.

analogous to the experiments executed at high-field. Notably, the nonlinear "masking" due to the resonant echo pulse is not observed here as the decoupling is performed with a hopping gate. The total transient phase pickup due to PIRS is reduced compared to Fig. 2. This is likely a consequence of the low magnetic field environment, as either a g-factor change or electron displacement in the gradient will result in a smaller shift in qubit Larmor frequency. In any case, the shuttling gate avoids the transient effect of the microwave burst, and the magnitude of the effect would decrease as the distance between local

operations is increased, making a low-field shuttling implementation of single-qubit gates an appealing approach to reduce crosstalk and improve multi-qubit control.

## Architecture proposal

The design of on-chip magnets for EDSR control of extensible 2D spin arrays presents a design challenge: engineering controllability and spectral addressability with a small pitch requires nanoscale magnetic variation. With magnets or current-carrying wires that are an

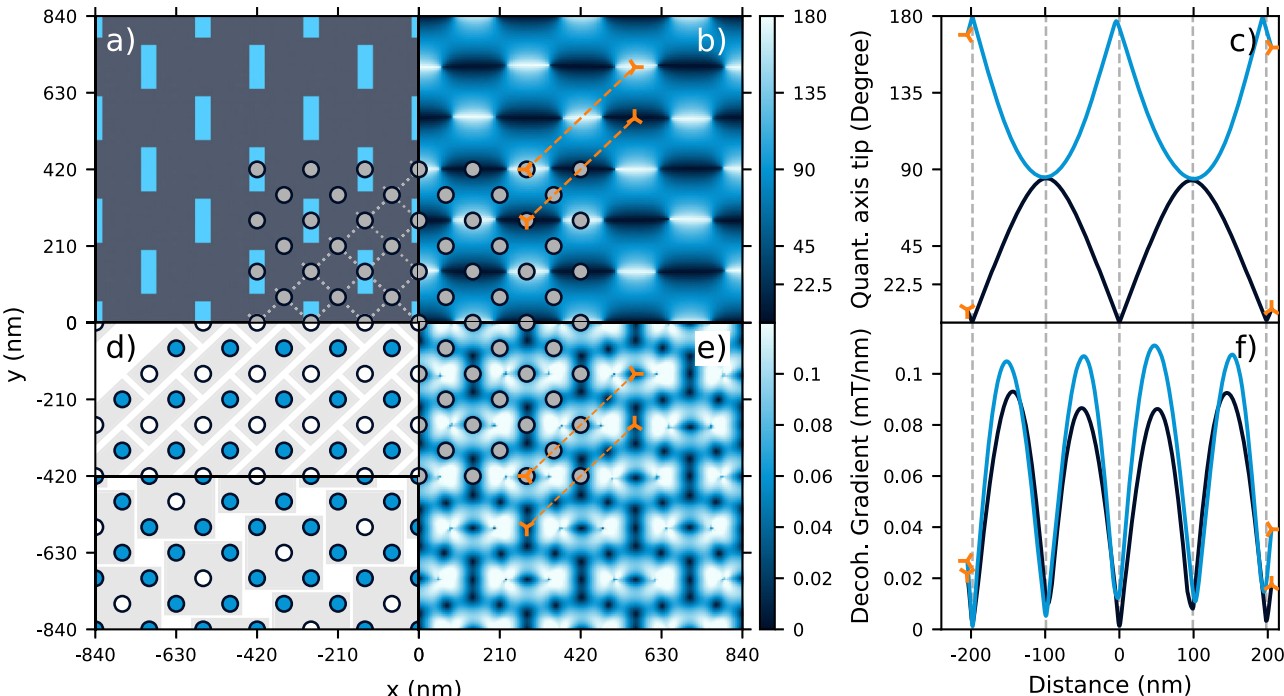

**Fig. 5 | Nanomagnet based tileable architecture utilizing hopping gates.**
**a** Zoomed in section of the proposed nanomagnet pattern with the magnetic material shown in bright blue. Each iron nanomagnet is 40 nm × 120 nm in area and 50 nm in out-of-plane height. In the lower right corner we show the envisioned quantum dot positions and their connectivity. These patterns extend across the contiguous (**a**), (**b**), (**d**), (**e**) and pertain to the data contained within them. **b** The relative change in quantization axis, assuming an isotropic g-factor, with respect to the central dot of the panel. The quantum well is located 100 nm below the bottom surface of the nanomagnet layer. The magnetic field is computed at zero external magnetic field after having allowed the nanomagnet magnetization to relax according to their shape anisotropy and material properties. **c** Line cuts corresponding to the orange markings in (**b**) illustrating the relative change in

quantization axis tips along two potential hopping axes. The observed asymmetries are a consequence of the finite mesh size used for simulation and interpolation artifacts. The gray dashed lines indicate the quantum dot positions. **d** Versatile filling of the quantum dot array: the bright blue filled dots indicate a quantum dot with an electron, the white markers indicate an empty quantum dot. The upper section shows a 50% filling, the lower section a denser filling with 80% occupancy. The gray shaded areas indicate the repeated unit cell. **e** The calculated decoherence gradient resulting from the nanomagnet stray field as described in (**b**). **f** Line cuts corresponding to the orange markings in (**e**) illustrating the decoherence gradient along two potential hopping axes. Minor artifacts from the finite mesh size and field interpolation are visible. Additional data is shown in Supplementary Fig. 2.

order-of-magnitude larger than the quantum dots themselves, this is difficult to achieve. Proposals making use of nanomagnets, with features similar in size to a ≈ 100 nm dot pitch, are a promising solution[18,19,32–35].

Inspired by these ideas, nanomagnets may also be used for hopping control. Compared to the microscopic g-tensor variations in germanium quantum dot arrays and similar platforms, local magnetic field variations could be engineered more deterministically. It is therefore worthwhile to revisit the design requirements for on-chip nanomagnets with hopping control in mind. Two specifications stand out compared to EDSR. First, qubit addressability with hopping gates is more localized and does not rely on the spectral separation of qubit frequencies. Second, the transverse gradient required for EDSR control effectively mandates a total magnetic field magnitude on the order of tens of mT, whereas tips in the quantization axis for hopping control (i.e. inhomogeneity of the magnetic field vector) are possible at more modest magnetic fields as illustrated by this work.

Exploiting these features of hopping control, we propose a periodic nanomagnet design as depicted in Fig. 5a. After applying an external magnetic field along the y-axis, the shape anisotropy imposed by the 40 nm × 120 nm × 50 nm iron magnet geometry causes the magnetization to persist even at ambient conditions[32]. The relaxed magnetization, simulated by the OOMMF software package[36] (see Supplementary Note 3), generates an inhomogeneous magnetic field that enables hopping conditions in a quantum well situated about 100 nm below the magnetic layer. Owing to the inhomogeneous magnetic vector field, a predictable pattern of quantization axis tips

(relative to a fixed reference point) of around 90 deg occurs with a periodicity of 200 nm as shown in Fig. 5b. Figure 5c shows this smoothly-varying tip along two shuttling axes in which hopping gates may be implemented between adjacent sites with a pitch of about 100 nm.

A sparse 2D array of silicon quantum dots may therefore be populated with flexible occupancy to realize different spin connectivity within the magnetic landscape. Figure 5d (upper) illustrates an instance where many dots may be left vacant to facilitate not only hopping gates but also shuttling to peripheral sensing regions or to implement two-qubit gates between distant spins. Figure 5d (lower) depicts a denser array where all qubits still have access to single-qubit logic via hopping. As any quantization axis tip between 45 deg and 135 deg allows Hadamard gates to be performed with a single shuttling cycle, this architecture is robust to some variation in both the placement/shape of the magnets and formation of the dots. As the nanomagnets can be fabricated over a large area, edge effects are relegated far beyond the extent of any finite quantum dot array. Of course, such a periodic array would also have to grapple with gate patterning and fanout as any other dense 2D array of dots would, but the issue of qubit controllability would not compound the difficulties. Provided such a fanout can accommodate the sparse magnetic layer, the design space for potential architectures becomes very large.

The nanomagnet array presented here confers two additional benefits beyond enabling hopping control that can also be leveraged in other potential designs. First, the qubit frequencies are distributed in a range below 400 MHz without any applied external field. This removes

the need for a solenoid in the cryogenic setup provided the magnets can be polarized at room temperature. Second, a periodic pattern of decoherence sweet spots also emerges to form a natural template upon which to place idling spins as seen in Fig. 5e. The proposed shuttling channels plotted in Fig. 5f show that even the maximum decoherence gradient encountered in the quantum well is not substantially larger than typical EDSR addressability gradients that range from 0.05 mT/nm to 0.1 mT/nm. We note that different combinations of ferromagnetic materials and nanomagnet geometries may be used to achieve similar outcomes.

The hopping approach is not without drawbacks. High fidelity hopping control requires voltage pulses with very precise timing on the order of tens of picoseconds and mandates working with physical phase gates as opposed to virtualized phase tracking in a rotating frame. In the case of silicon, the microscopic valley degree of freedom may interfere with charge shuttling in cases where the valley phase difference between dots modifies the effective tunnel coupling[30,37–39]. Also, the nanomagnets compete for space with the fanout of the gate electrodes that define the dots. Beyond engineering for single-qubit control, considering Zeeman energy differences for high-fidelity two-qubit gates[40] and Pauli spin blockade readout[10] would be necessary to demonstrate a viable architecture. Engineering locations with zero magnetic field could also be useful for parking spins to alleviate the task of phase-tracking while idling qubits. Although we do not anticipate anything inhibiting these features with nanomagnet designs, we leave their details for future studies.

## Discussion

We have used a $2 \times 2$ $^{28}$Si/SiGe quantum dot device to explore two ways in which future 2D arrays of single-electron spins in silicon may be controlled. Using microwave driving and barrier gate voltage pulses, we demonstrated universal control over a four-qubit system using conventional strategies from 1D arrays with only mild alterations. The performance was primarily limited by poor exchange control owing to the weak tunability of nearest-neighbor tunnel couplings. One potential remedy would be to modify the geometry of the barriers and invert the gate layers in fabrication to increase the lever arm controlling exchange. Another possibility would be the inclusion of a central gate to facilitate the decoupling of spins without requiring such large pulse amplitudes. Regardless, this demonstration of a 2D silicon spin array shows that established qubit control methods are sufficient for scaling beyond linear architectures. Longer bilinear arrays and perhaps trilinear devices ought to be compatible with established micromagnet designs. However, we also found that resonant single-qubit control causes crosstalk that complicates multi-qubit operation.

By opposing the polarization of the micromagnet with the external field to create a net vector field that varies over the scale of the array, we implement hopping gates that allow for baseband control of single spin qubits. Our initial characterization indicates several advantages of this operating regime. First, the baseband control pulses do not impart the transient phase crosstalk that was observed with resonant control. This may be attributed to the reduced power dissipation per logical operation which in turn reduces local heating effects[25]. Second, the reduced magnetic field reduces the coupling strength to charge noise thus increasing the Hahn echo coherence substantially. Whereas the device was not optimized for hopping gate operation, we already demonstrate a hopping gate that is competitive with the EDSR control demonstrated in this device both in terms of speed and fidelity. We believe the same aforementioned improvements in tunnel coupling control would permit better fidelities in future implementations, and the gate speed would effectively double by increasing the tip of the quantization axis between dots to above 45 deg.

Finally, we explore how engineering on-chip magnets for hopping spins alleviates many of the challenges that impede scaling EDSR

control. We put forward an illustrative example of a periodic nanomagnet pattern that creates a deterministic pattern of decoherence sweet spots and nearest-neighbor tip angles for implementing hopping spins in an arbitrarily large 2D array. These conditions can further enhance coherence times, increase the single-qubit gate speed, and do not require a superconducting solenoid to provide an external field. We believe further investigation into nanomagnet designs for hopping spin control would be very fruitful.

## Methods

### Device design and fabrication

The device layout used in this work are similar to the ones used in our previous work[26]. The heterostructure comprises a $Si_{0.69}Ge_{0.31}$ strained relaxed buffer (SRB), a 7 nm tensile-strained $^{28}$Si quantum well (isotopic enrichment 800 ppm), and a 30 nm $Si_{0.69}Ge_{0.31}$ barrier passivated by an amorphous Si-rich layer[41]. Substantial noise from the magnet power supply in driven mode prevented a reliable magnetospectroscopic measurement of the valley splittings in this device. The average valley splittings of 240 $\mu$eV measured in ref. [41] are representative of what we would expect in this device, though it is possible that local microscopic disorder may yield outliers for particular quantum dots.

On top of the heterostructure we fabricate a multilayer gate stack isolated by a native $SiO_x$ layer and a 10 nm thick layer of $AlO_x$ deposited using atomic layer deposition (ALD). The first metallic layer (3:17 nm Ti:Pd) connects to the ohmic contacts and defines screening gates. The screening gates define the area that will host the four quantum dots and two SETs and prevent electron accumulation or spurious dot formation in the fan out. Furthermore, the screening gates confining the $2 \times 2$ array are designed as coplanar waveguides to effectively deliver microwave signals for EDSR control. The second layer (3:27 nm Ti:Pd) includes all plunger and accumulation gates, while the third layer (3:27 nm Ti:Pd) contains all barrier gates. The top layer of the gate stack (3:200 nm Ti:Co) contains only the micromagnet. Each layer is electrically isolated from the ones before by 5 nm $AlO_x$ grown with ALD.

The SETs are measured using RF reflectometry implemented using the split-gate approach[42]. The size of the accumulation gates was substantially increased compared to[26] to increase the capacitance to the 2D electron gas (2DEG). The split accumulation gate could be depleted to create a high-resistance path from the 2DEG to the ohmic to prevent signal leakage.

The fan out of the barrier and plunger gates is shown in Supplementary Fig. 7. The starred connections indicate the bond pads corresponding to B43, P3 and B23 which have uniquely circuitous traces to adapt to the distribution of fast-lines on the sample PCB. We consistently observe readout degradation on a timescale at least as long as a 10 $\mu$s single-shot measurement when applying baseband pulses to these gates. As the cross-capacitance between these traces and adjacent bondpads (many of which connect to unvirtualized gates) is much larger in the fan out than at the scale of the quantum dot array, we suspect their suboptimal routing is correlated with the drop in readout quality and that these problems can be alleviated in future designs.

### Initialization, control, and readout

A single-shot measurement includes initialization, manipulation and readout. Initialization and readout are performed using Pauli spin blockade and post-selection similar to[9]. While PSB could be achieved in 4 qubit pairs in this device, we focus on pairs (4,1) and (2,3) as they offered superior performance. All shots begin with a round of parity-mode PSB measurement where parallel spin states are blockaded in the (1,1) charge state and antiparallel spin states are permitted to tunnel to the doubly-occupied singlet state S(2,0). A fast diabatic ramp is used to pulse from the readout point to the (1,1) charge state. When the post-selected initial state is S(2,0), the fast ramp yields a mostly-

entangled state (see Supplementary Fig. 3) with consistently higher single-qubit visibility than what is obtained with an adiabatic ramp to the (1,1) charge state. Prior to measurement, a wait-time of 1μs to 5μs is included to permit both antiparallel spin states to decay to the S(2,0) state. The charge sensor signal is integrated for 20 μs. Some datasets are post-processed to shift the readout threshold to an optimal point.

While a longer integration time improves readout quality somewhat, there appear to be other fundamental limits to the qubit visibilities we can achieve. One factor is the limited tunnel coupling tunability in this particular device upon which the PSB mechanism depends very sensitively. At the high-field operating condition, the spins' quantization axes are estimated to be aligned within a few degrees. However, at low-field operation the expected tip between quantization axes in the PSB pair is estimated to be about 20 deg. This causes an opening of the S(2,0)-$|\downarrow\downarrow\rangle$ avoided crossing which may impede a linear adiabatic ramp from the S(2,0) state to the $|\downarrow\uparrow\rangle$ for high-fidelity initialization. Pulse-shaping and active feedback may be used to mitigate this, but these were not investigated in this work. Alternatively, an adiabatic ramp through a sizeable S(2,0)-$|\downarrow\downarrow\rangle$ avoided crossing may permit initialization directly into the $|\downarrow\downarrow\rangle$ state as is often done with hole spin qubits. In the future, spin qubit designs could make use of separate zones for readout and control, which are individually optimized by local nanomagnets, for example.

Qubit manipulation includes resonant drive and diabatic spin shuttling for single-qubit gates. The Hamiltonian for the single-qubit gates are discussed in Supplementary Notes 1 and 5 respectively. All microwave pulses are padded with 5 ns wait times to allow for a transient ring-down of the microwave source. Furthermore, the IQ amplitudes and phases are adjusted for each qubit at its center-frequency to suppress LO-leakage, unwanted sidebands and other spurious tones during IQ modulation. In all cases, a square pulse shape is used for the I and Q waveforms.

The exchange interaction, given by $H_{exch} = 2\pi\hbar J \mathbf{S} \cdot \mathbf{S}$, is turned on and off via square barrier pulses for the DCZ oscillations shown in Fig. 1c. Due to the large Zeeman energy difference present between all qubit pairs, we take the two-qubit unitary evolution to be $U_{CZ}(t) = \text{diag}(1, \exp(i\pi Jt), \exp(i\pi Jt), 1)$.

The presented single- and two-qubit experiments were run at a wide variety of temperatures and fields. The chevron patterns in Fig. 1b and the randomized benchmarking in Fig. 4a were run at 250 mK and 200 mT to prevent heating effects. The decoupled CZ oscillations were measured at the same field but at base temperature (around 20 mK). All external field sweeps and shuttling gate experiments were conducted at base temperature.

## Randomized benchmarking

In this work, we benchmark single-qubit gates implemented via resonant control and hopping control. Here we discuss the implementation of the Clifford gate sets in greater detail to interpret the average gate fidelities that are extracted for both gate flavors. For standard randomized benchmarking, we extract fidelities by fitting the decay curve to:

$$y(N) = A + B(1-p)^N, \tag{1}$$

where $N$ is the number of Clifford gates in the RB sequence, and $A$, $B$, and $p$ are fitting parameters. The estimated Clifford gate fidelity is then estimated as $1 - p/2$. The reported error bars correspond to the standard deviations acquired from curve fitting.

When using resonant control, the X/Y compilation is used (see Table II of ref. 43) whereby all Clifford gates are compiled into ±90 deg rotations about the X and Y axes of the Bloch sphere. A single microwave burst operation is calibrated: it has a constant amplitude, frequency, duration, and a post-burst padding of 5 ns to allow for the microwave source to ring down. The only difference between pulses is the relative phase with which the I and Q modulation channels begin.

For a given sequence of Clifford operations, the error contributed by each physical gate should be nominally identical in the absence of any non-Markovian effects. As the X/Y compilation of the 24 Cliffords contains 52 primitive gates in total, the average resonant gate fidelity $F_{avg}^{res}$ from the set $\{X_{\pm90}, Y_{\pm90}\}$ is estimated to relate to the resonant Clifford gate fidelity $F_{Clif}^{res}$ as $F_{avg}^{res} = 1 - (1 - F_{Clif}^{res}) \cdot \frac{24}{52}$. For $Q_1$ to $Q_4$ respectively, the Clifford gate fidelities are measured to be $F_{Clif}^{res} = \{99.28(5)\%, 99.58(4)\%, 98.7(2)\%, 98.5(3)\%\}$. We use 20 random Clifford circuits per data point, and average the result of about 1200 post-selected single shot measurements per circuit. The corresponding average resonant gate fidelities are, respectively, $F_{avg}^{res} = \{99.67(2)\%, 99.80(2)\%, 99.39(8)\%, 99.3(2)\%\}$.

When using hopping control, the X/Z compilation is used (see again Table II of ref. 43) as the gate set is comprised of a unique $X_{90}$ hopping operation as well as Z rotations implemented with physical idling times. Negative rotation about the X axis is implemented by a $Z_{180}$ prior to a $X_{90}$ hopping gate. In contrast to the resonant gate compilation, the X and Z operations are of a fundamentally different nature and therefore contribute to the error rate in distinct ways. First, $X_{90}$ operations require more time than Z rotations and therefore pick up more error through dephasing. Second, $X_{90}$ operations require four precise intervals of Larmor precession whereas Z rotations require only one. This means that any uncorrelated jitter in the AWG ramps will cause more error in the $X_{90}$ operations. Third, as $X_{90}$ operations involve shuttling, errors may arise due to imperfect charge or spin transfer between dots. Therefore, $X_{90}$ gates should be more prone to error than any Z rotation.

Interleaved randomized benchmarking data was not taken to directly estimate individual gate fidelities, so we propose two ways of interpreting the fidelity of the individual physical operations. From randomized benchmarking, we obtain a Clifford gate fidelity $F_{Clif}^{hop}$. We use 250 random Clifford circuits per data point, and average the result of about 1500 (200 circuits) and 750 (50 circuits) post-selected single-shot measurements per circuit. The X/Z Clifford set is compiled from 89 physical gates in the set $\{X_{\pm90}, Z_{\pm90}, Z_{180}\}$, so one may estimate the average gate fidelity of this set as $F_{avg}^{hop} = 1 - (1 - F_{Clif}^{hop}) \cdot \frac{24}{89}$. Alternatively, one may estimate a lower bound of the $X_{90}$ gate fidelity by attributing all of the error to the $X_{90}$ operations on the basis that all Z operations are effectively brief extensions to the preceding $X_{90}$ sequence. As each Clifford gate in the X/Z compilation contains exactly 2 $X_{90}$ gates, the $X_{90}$ fidelity may be bounded as $F_{X_{90}}^{hop} = 1 - (1 - F_{Clif}^{hop}) \cdot \frac{1}{2}$. We run standard randomized benchmarking of the baseband gate set after initializing both in the odd and even parity states as shown in Fig. 4f. To be conservative in our evaluation of the X90 hopping gate quality, we extract the Clifford gate fidelity of $F_{Clif, odd}^{hop} = 99.01(11)\%$ for $Q_1$ from the odd parity decay. It follows that $F_{avg, odd}^{hop} = 99.73(3)\%$ and $F_{X_{90}, odd}^{hop} = 99.50(6)\%$. This latter value is the estimated fidelity bound reported in the abstract. For the even parity branch we fit $F_{Clif, even}^{hop} = 99.49(7)\%$, $F_{avg, even}^{hop} = 99.86(2)\%$ and $F_{X_{90}, even}^{hop} = 99.74(4)\%$. The even parity branch contains a small oscillatory artifact that persists even with substantial averaging, and we are unsure of the origin of this artifact.

To gain insight about possible state leakage during hopping operation, we also employ the fitting method of blind randomized benchmarking (blind RB)[31]. Blind RB was developed to estimate the influence of leakage on the operation fidelity within an encoded subspace. We can use a similar interpretation here to estimate the role of leakage for the hopping gate. Blind RB makes use of two separate RB decay curves: one, $y_0(N)$, where the inverting Clifford is calculated to return the qubit to the initial state, and another, $y_1(N)$, where the inverting Clifford is modified to result in a net qubit-flip. Our implementation is imperfect in the sense that the even-parity decay curve was prepared with a physical $X_{180}$ gate that introduces additional SPAM errors compared to the odd-parity decay curve, but this effect should be small. The two RB curves can be fit to the following system of

equations:

$$y_0(N) = A + B(1-p)^N + C(1-q)^N \qquad (2)$$

$$y_1(N) = A - B(1-p)^N + C(1-q)^N, \qquad (3)$$

where $N$ is the number of Clifford gates in the RB sequence, and $A$, $B$, $C$, $p$, and $q$ are fitting parameters. The estimated SPAM-normalized leakage rate per Clifford operation is estimated as:

$$\Gamma = -Cq/B, \qquad (4)$$

the total SPAM-normalized error rate per Clifford operation is estimated as:

$$\epsilon = p/2 - Cq/2B, \qquad (5)$$

and the error rate within the qubit subspace is inferred to be $\epsilon_q = \epsilon - \Gamma$ (see ref. 31 for derivations). For the hopping gate benchmarking data shown in Fig. 4f, blind RB yields a leakage rate of $\Gamma = 0.13(5)\%$, a total error rate of $\epsilon = 0.74(5)\%$, and a qubit error rate of $\epsilon_q = 0.61(5)\%$ for a Clifford gate fidelity estimate of $1 - \epsilon = 99.26(5)\%$ which is comparable to the estimates of the standard RB fits. We discuss this observed leakage in Supplementary Note 7.

We note that neither the X/Y or X/Z compilations used are maximally efficient at compiling Clifford operations from primitive operations. Therefore, the Clifford gate fidelities extracted from randomized benchmarking could themselves be improved by tailoring the gate set to the physical gates. The average gate fidelity estimates should therefore be the most reasonable quantitative benchmark with which to compare the gate performance.

### Tip angle fitting

There are multiple strategies to estimate the quantization axis tip $\theta$ between two dots. The visibility of oscillations observed after diabatically shuttling from one dot to another can be directly related to the tip in quantization axis[24]. This provides a reasonable estimate if the maximum visibility of oscillations can be established and provides good contrast. Alternatively, the Larmor frequency as a function of detuning across the hopping double-dot pair can be measured and fit to Supplementary Eq. (28)[24,25]. Both of these methods are made difficult in our case by not having EDSR control at the magnetic field condition at which hopping gates are possible.

Here, we take advantage of the fact that a number of repetitions $r$ of the shuttling sequence as drawn in Fig. 4a will result in a unique 2D "fingerprint" as a function of the timing parameters $t_1$ and $t_2$. Generally, the pattern will depend more sensitively on the tip $\theta$ as $r$ increases. Exemplary patterns for both dot pairs are shown in Supplementary Fig. 4.

We may model the pattern using the two-level Hamiltonian given in Supplementary Eq. (30) and make the approximation that the change in the Hamiltonian as a function of interdot detuning is perfectly sharp given the nanosecond-scale detuning ramp times[25]. The time evolution is therefore a product of unitary Larmor precessions. In the initial dot, this precession occurs at an angular frequency of $\omega_{\text{init}}$ about a quantization axis $\hat{z} = (0, 0, 1)^T$. In the tipped dot, this precession occurs at an angular frequency of $\omega_{\text{tip}}$ about a quantization axis $\hat{\theta} = (\sin\theta, 0, \cos\theta)^T$. Since the detuning ramp times are still finite, precession occurs during the ramps and can be accounted for by adjusting the nominal precession time such that $t_1' = t_1 - t_{1,\text{offset}}$ and $t_2' = t_2 - t_{2,\text{offset}}$. The unitary evolution for $r$ repetitions of the sequence shown in Fig. 4a is therefore given by:

$$U(t_1', t_2', t_{\text{add}}, \omega_{\text{init}}, \omega_{\text{tip}}, \theta) = \\ \left( R_z(\omega_{\text{init}}(t_2' + t_{\text{add}})) R_\theta(\omega_{\text{tip}} t_1') R_z(\omega_{\text{init}} t_2') R_\theta(\omega_{\text{tip}} t_1') \right)^r \qquad (6)$$

where $R_n(\alpha) = \exp(-i\alpha\hat{n} \cdot \vec{\sigma}/2)$ indicates a positive rotation about the unit vector $\hat{n}$ by an angle $\alpha$. The measured pattern is given by:

$$p_{\text{odd}}(t_1', t_2', t_{\text{add}}, \omega_{\text{init}}, \omega_{\text{tip}}, \theta, A, B) = \\ A\text{Tr}(OU\rho_{\text{odd}}U^\dagger) + B, \qquad (7)$$

where $\rho_{\text{odd}}$ is the initialized odd-parity state, $O = (\mathbb{1} - \sigma_z \otimes \sigma_z)/2$ is the odd-parity observable, and $A$ and $B$ are visibility and offset corrections due to SPAM errors.

We fit the experimental data to Eq. (7) with a pixel-by-pixel least-squares optimization over the parameters $t_{1,\text{offset}}$, $t_{2,\text{offset}}$, $\omega_{\text{init}}$, $\omega_{\text{tip}}$, $A$, $B$, and $\theta$. $t_{\text{add}}$ is known exactly from the experiment definition. As the qubit frequencies, offset times, and visibilities are well-estimated from other experiments, they are optimized within narrow bounds. For $r = 4$, the optimized value for $\theta$ is accurate within a degree. For fewer repetitions, the fit is less accurate, and we instead simulate using the $r = 4$ fit directly.

## Data availability
Data and scripts used in this publication are available in the Zenodo repository[44].

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

## Acknowledgements

We thank M. Aldeghi, C.-A. Wang, H. Tidjani, and I. Fernandez de Fuentes for very helpful discussions, L. Tryputen and S. V. Amitonov for fabrication and design input as well as taking the presented SEM image, and O. Benningshof for technical support. Furthermore, we thank other members of the Vandersypen, Veldhorst, and Scappucci groups for their input and feedback. We acknowledge financial support from Intel, the Army Research Office (ARO) under grant number W911NF-17-1-0274 and W911NF2310110, and the Dutch the Ministry for Economic Affairs through the allowance for Topconsortia for Knowledge and Innovation (TKI). We also acknowledge support from the "Quantum Inspire-the Dutch Quantum Computer in the Cloud" project (Project No. NWA.1292.19.194) of the NWA research program "Research on Routes by Consortia (ORC)," which is funded by the Dutch Research Council (NWO).

## Author contributions

F.K.U. and B.U. performed the experiments and data analysis with input from Y.M, O.P.-C., and M.M. Micromagnet simulations were carried out by B.U. Nanomagnet simulations were carried out by E.R. The sample was designed by F.K.U. with input from A.S.I., and M.M., and fabricated by S.K. on a heterostructure grown by A.S and G.S. Control libraries for the experiment were developed by S.L.S. F.K.U., B.U., and L.M.K.V. wrote the manuscript with input from all authors. M.V. and L.M.K.V. supervised the project.

## Competing interests

The authors F.K. Unseld, B. Undseth, E. Raymenants, and L.M.K. Vandersypen are inventors on a patent application on nanomagnet designs for hopping spin control filed by Delft University of Technology under the application number NL2039255. The remaining authors declare that they have no competing interests.
