## [Transparent Peer Review file · Nature Communications]

Baseband control of single-electron silicon spin qubits in two dimensions

Corresponding Author: Mr Florian Unseld

Version 0:

Reviewer comments:

Reviewer #1

(Remarks to the Author)

The manuscript "Baseband control of single-electron silicon spin qubits in two dimensions" is an experimental demonstration of hopping-mode spin qubits in silicon that includes a comparison to resonantly driven single-spin qubits in the same device.

The authors argue that the appeal of this control scheme is that it utilizes baseband control pulses as opposed to RF pulses, which avoids some of the crosstalk and heating issues that are identified as key challenges for the resonantly driven approach. The first demonstration of this style of hopping qubit was in Germanium hole spins, which could rely on the naturally occurring g-factor anisotropy to give large enough axis tilt between sites to accomplish this pulse scheme. Silicon does not have this same g-factor anisotropy and therefore must rely on an engineered approach to create the required magnetic field profile, demonstrated here. To bolster the argument, the authors explore a proposed design for generating the correct magnetic field profile theoretically and show that baseband control of single-electron spins in silicon may have a path forward.

The experiments in the paper are well designed as a comparison between EDSR controlled qubits and baseband hopping qubits within the same device. The PIRS effect for EDSR qubits, studied in section 2 of the paper, has been documented in other papers and still eludes an explanation here. The fact that PIRS is shown not to be so detrimental for hopping qubits helps bolster the argument for the baseband qubit.

It seems a better device would be needed to push the limits of this control scheme and gain an understanding of subtle error mechanisms, but as an initial demonstration it is well designed, timely, and relevant for the field. It should be appropriate for Nature Communications after some questions are addressed.

Questions for the authors:

1. What is the reason for the low EDSR fidelities compared to state of the art shown by this group? Is the 2MHz rabi frequency attainable here (before observing overdriving) too low compared to dephasing rates?
2. The valley states in silicon may play a role as an error or leakage pathway for protocols that involve interdot charge transfers, which is discussed theoretically in their citation #36 Ginzal et al. The authors did not report valley splittings for these dots, but if they have the data, it should be included in these results.
 - a. Previous results from this group were gathered from a Si/SiGe heterostructure that was reported to have large valley splittings [citation #41 Esposti et al.], were these from similarly grown heterostructures with the expectation of having large valley splittings?
3. The RB data presented in figure 4f looks somewhat like the RB data from the exchange only qubits which have a significant leakage contribution to their error. Does this measurement truly asymptote at ~78% like the model fit here is suggesting? Have the authors considered a model like that of Nature Nanotechnology volume 14, pages747–750 (2019)? In which case, there are ways of quantifying the leakage term and the gate error terms separately to better understand the system.
4. Supp. Figure S1 seems to have horizontal bands of higher and lower coherence. For instance, in panel A there is a rather incoherent stripe centered around 10ns ramp time, then a coherent section around 22ns. Then the coherence decreases again and revives around 42ns. Is this a measurement artifact? Do you observe this in repeated datasets?
 - a. What ramp times were used for the hopping gates in the RB experiment? Did they correspond to the "best looking" regions in Fig. S1 data? Perhaps I am overlooking it, but this number should appear in the main text for the reader's reference.
 - b. The value for the tunnel coupling used during the hopping gate experiments should appear in the main text in the paragraph that starts on line 317.

5. The measurement fidelity in this manuscript and the Ge hopping qubit papers is low compared to state of the art. Are there fundamental challenges to achieving higher measurement fidelity for this qubit encoding with tilted quantization axes and magnetic field gradients that the field needs to know about and solve or were the readout results here purely limited by technical challenges?

6. How long does it take for the B43, P3, B23 induced crosstalk on the sensor to settle out? Why do baseband pulses on P3 during the control sequence impact the readout portion which is several microseconds later? Would the same thing be observed on a charge sensor wired for low frequency conductance measurements instead of RF reflectometry? Does this have implications for scaling this readout approach?

7. Is the reduction in visibility for the rabi oscillations in Extended Data Fig. 2 due to a similar effect? Why is the visibility reduced with a bias towards odd parity measurement outcomes?

Reviewer #2

(Remarks to the Author)

Key results:

In this manuscript, Unseld et al. demonstrated that micromagnet-based EDSR control of single-electron spins, traditionally used in 1D arrays of gate-defined quantum dots (QDs), is also feasible in a small 2D QD array in silicon. They confirmed this in a 2x2-QD array, additionally showcasing nearest-neighbor exchange control. They benchmarked the performance of the four-qubit system, particularly in relation to off-resonant drive-induced crosstalk. Furthermore, by utilizing the same system as a two-qubit setup, they demonstrated qubit operation via hopping gates enabled by engineered magnetic stray fields. They also analyzed the qubit coherence, fidelity, and crosstalk associated with the hopping gate. They obtained high gate fidelities with both the approaches: a lower bound of 99.50% for the hopping gate and an average one of 99.54% for the resonant gate. Finally, they proposed a nanomagnet pattern to enable hopping control across an arbitrarily large 2D array.

Validity:

The validity and robustness of the data interpretation show no flaws, and the derived conclusions are sound.

Significance:

The exploitation of EDSR in a 2x2 Si QD array of single-electron spins, as reported in this manuscript, represents a pioneering yet crucial step toward the application of EDSR in scaled-up systems. Despite the challenges posed by microwaves—particularly heating effects and PIRS—which are likely to remain limiting factors, the introduction of hopping gates offers a promising alternative. Hopping gates eliminate the need for microwaves, thereby mitigating heating effects and enhancing scalability. The engineered micromagnets, with their specific design presented in this work, show significant potential in enabling hopping gates in larger 2D arrays, opening new possibilities for future advancements in quantum computing in silicon.

Data and methodology:

The validity of the approach is strong, and the quality of the presentation is excellent. Moreover, the supplementary notes are very helpful in providing explanations for the experimental results from a theoretical/modeling perspective. The quality of the results is very high, with only a minor weakness in qubit visibility for the hopping gates shown in Figure 1c. However, the authors have already provided a satisfactory explanation for the degraded readout signal in Section V of the manuscript.

Analytical approach:

The analytical approach is consistently strong throughout the manuscript, both in the resonant gate and hopping gate sections.

Suggested improvements:

Here are some minor comments on the Extended Data Figures:

I would include the colorbar for the experimental (and fitted) qubit visibility in Figures 3 and 4. Moreover, I recommend adding the corresponding colorbar(s) to the graphs including the simulated magnetic fields shown in Figures 9(a), (c), (e), and (g).

Clarity and context:

The text is extremely clear and accessible, and the results are provided with great context and consideration of previous work.

References:

The manuscript appropriately references both previous and state-of-the-art literature.

Version 1:

Reviewer comments:

Reviewer #1

(Remarks to the Author)

The authors addressed my concerns, and the revised manuscript is stronger as a result and ready for publication. I

appreciate the additional analysis on leakage and the comparison between tuning qubits at 18 ueV and 48 ueV tunnel couplings. It helps one develop an error budget for the qubit. The body of work feels more complete and is a good start for the hopping spin qubit in silicon.

I recommend the authors consider a valley spectroscopy routine like detuning axis pulsed spectroscopy in the future to probe valley splitting in the single electron regime with a protocol that closely approximates the shuttling protocol. It would be important for this encoding scheme going forward to study how valleys contribute to leakage. For now, I think not having the valley data is ok.

Reviewer #2

(Remarks to the Author)

I thank the authors for submitting a revised manuscript that addresses my minor suggestions as well as those raised by the other reviewer. I confirm that the article is suitable for publication in Nature Communications.

Reviewer #1 (Remarks to the Author):

The manuscript “Baseband control of single-electron silicon spin qubits in two dimensions” is an experimental demonstration of hopping-mode spin qubits in silicon that includes a comparison to resonantly driven single-spin qubits in the same device. The authors argue that the appeal of this control scheme is that it utilizes baseband control pulses as opposed to RF pulses, which avoids some of the crosstalk and heating issues that are identified as key challenges for the resonantly driven approach. The first demonstration of this style of hopping qubit was in Germanium hole spins, which could rely on the naturally occurring g-factor anisotropy to give large enough axis tilt between sites to accomplish this pulse scheme.

Silicon does not have this same g-factor anisotropy and therefore must rely on an engineered approach to create the required magnetic field profile, demonstrated here. To bolster the argument, the authors explore a proposed design for generating the correct magnetic field profile theoretically and show that baseband control of single-electron spins in silicon may have a path forward.

The experiments in the paper are well designed as a comparison between EDSR controlled qubits and baseband hopping qubits within the same device. The PIRS effect for EDSR qubits, studied in section 2 of the paper, has been documented in other papers and still eludes an explanation here. The fact that PIRS is shown not to be so detrimental for hopping qubits helps bolster the argument for the baseband qubit.

It seems a better device would be needed to push the limits of this control scheme and gain an understanding of subtle error mechanisms, but as an initial demonstration it is well designed, timely, and relevant for the field. It should be appropriate for Nature Communications after some questions are addressed.

We thank the reviewer for their time in reviewing our manuscript and are happy to see the positive recommendation. Below we respond point-by-point to the comments brought up by the referee.

Questions for the authors:

1. What is the reason for the low EDSR fidelities compared to state of the art shown by this group? Is the 2MHz rabi frequency attainable here (before observing overdriving) too low compared to dephasing rates?

We believe the errors are predominantly limited by the T_2 Rabi decay of the qubits when subject to the driving field compared to the quasi-static noise fluctuations associated with

the measured T_2^* dephasing. We have updated the manuscript to emphasize this from line 160, and we wrote an additional supplementary section to explain our reasoning in detail. We estimated the expected infidelity due to off-resonant driving because of quasi-static noise based on our measured T_2^* and find error rates on the order of 0.05%. We also estimated the expected infidelity due to the T_2 Rabi-limited quality factors of the driven qubits and find an error rate an order of magnitude higher, about 0.7%. This latter value approximately agrees with the 0.5% error rates we observed from randomized benchmarking. When performing this analysis, we also realized that we had originally fitted the T_2 Rabi using a Gaussian decay, whereas an exponential decay is more suitable. Our estimates for the quality factors have therefore been increased.

Beyond the data included in Extended Data Figure 2, we did not characterize the quality factor or fidelity of EDSR at higher Rabi frequencies. This was due to our choice at the time to focus on the PIRS effect across all 4 qubits rather than optimize the qubits individually, and the power required for moderate Rabi frequencies was already sufficient to see this effect. Previous work focused on optimizing the quality factor of EDSR has found that increasing the Rabi frequency improves the fidelity up to the point where the side-effects (e.g. heating) limit the driving quality (Noiri, et al. 2022). We would predict that the faster qubits (e.g. Q2) could achieve a proportionally higher quality factor by increasing the Rabi frequency, but we do not have the data to quantify this.

2. The valley states in silicon may play a role as an error or leakage pathway for protocols that involve interdot charge transfers, which is discussed theoretically in their citation #36 Ginzel et al. The authors did not report valley splittings for these dots, but if they have the data, it should be included in these results.

We did spend considerable time attempting to characterize the valley splittings through magnetospectroscopy. However, in the setup used for these experiments, the superconducting magnet power supply couples substantially to the readout chain. Although this is not a problem when operating the magnet in persistent mode, for sweeping the magnetic field it substantially lowers the SNR. Through the noise, we unfortunately were not able to make a confident estimate of the valley splittings in these quantum dots. We have added this comment in line 870 of the Methods.

a. Previous results from this group were gathered from a Si/SiGe heterostructure that was reported to have large valley splittings [citation #41 Esposti et al.], were these from similarly grown heterostructures with the expectation of having large valley splittings?

Indeed, the wafer used to fabricate the 2x2 device discussed in this manuscript is from the same process as reported in citation #41. In line 870 of the Methods, we now state explicitly that the average valley splittings in #41 are representative example of what we would expect in this 2x2 device, with the caveat that it is still possible that lower valley splittings are possible due to local disorder.

3. The RB data presented in figure 4f looks somewhat like the RB data from the exchange only qubits which have a significant leakage contribution to their error. Does this measurement truly asymptote at ~78% like the model fit here is suggesting? Have the authors considered a model like that of Nature Nanotechnology volume 14, pages747–750 (2019)? In which case, there are ways of quantifying the leakage term and the gate error terms separately to better understand the system.

We agree with the reviewer’s suggestion and believe the paper is improved with an additional discussion of leakage supported by blind RB. We have added the result of fitting the data of Figure 4f to the main text at line 391 and adjusted our wording based on the results. Blind RB suggests a leakage rate of about 0.1% and a total error rate of 0.7% in the gate. This result suggests that the gate fidelity in this case is more limited by other sources such as calibration errors and stochastic noise. However, we do not have RB data extending beyond 1000 Clifford gates to see the full asymptote of potential leakage.

To add additional context to the discussion of leakage, we have also included a short supplementary section using blind RB to fit to an earlier calibrated gate when the tunnel coupling was substantially lower (about 18 ueV as opposed to about 48 ueV) and the leakage rate in that case is fitted to be about 1%. We speculate that the restrictive tunnel coupling increases the chance of exciting the charge state into a higher valley-orbit, therefore lifting PSB and causing leakage that appears as an odd-parity spin state.

Although the blind RB fit leads to a moderately improved hopping Clifford gate fidelity (99.3%) compared to standard RB (99.0%), we have chosen to keep our estimated hopping gate fidelity quoted in the abstract based on the more conservative fit to standard RB.

One caveat about our analysis is that the conceptual X180 gate used to observe the even parity RB decay was implemented physically and not incorporated into the inverting Clifford. This oversight means that the measurement ought to have a systematically higher error rate than the standard decay, but likely the effect is relatively small. We have extended the description of randomized benchmarking in the Methods to include the relevant equations.

4. Supp. Figure S1 seems to have horizontal bands of higher and lower coherence. For instance, in panel A there is a rather incoherent stripe centered around 10ns ramp time, then a coherent section around 22ns. Then the coherence decreases again and revives around 42ns. Is this a measurement artifact? Do you observe this in repeated datasets?

This was puzzling to us at the time and remains so. Indeed, we did observe a similar effect across multiple data sets. We also observed that the hopping pattern of Figure 4d) would be suppressed for ~ 12 ns ramp times. The incoherent and coherent stripes at ~ 10 ns and 22 ns ramp times respectively is a bigger mystery that we can only speculate on without more detailed knowledge of the microscopic valley-orbit structure of the dots. It may be that during interdot tunneling, the state does not cleanly transfer adiabatically or diabatically between spin-valley eigenstates, and therefore some phase is picked up that destructively interferes for particular ramp/wait timing combinations. A simplified model ignoring inter- and intra-valley tunneling was sufficient to provide a first-order understanding of the hopping gate, but we acknowledge that a more detailed model such as in references #36-38 would be required to verify or refute this hypothesis. We have added this clarification at line 218 of the Supplementary Material.

a. What ramp times were used for the hopping gates in the RB experiment? Did they correspond to the “best looking” regions in Fig. S1 data? Perhaps I am overlooking it, but this number should appear in the main text for the reader’s reference.

For the data presented in the main text and Extended Data figures, a nominal ramp time of 2 ns was used. This was determined to be the fastest nominal ramp time accurately achievable by the AWGs used in the experiment, and we aimed to ramp as quickly as possible to minimize decoherence during shuttling. We did not systematically study the gate fidelity as a function of the ramp time used. The experiment shown in Fig. S1 did, however, inform us of ramp times that did not work well. We have added the ramp time in line 351 of the manuscript.

b. The value for the tunnel coupling used during the hopping gate experiments should appear in the main text in the paragraph that starts on line 317.

We estimate the tunnel coupling used for the best hopping gate fidelity achieved in this work to be roughly 48 μeV based on our fitting in Extended Data Fig. 7. We have also added this value to line 351 of the modified manuscript. The patterns shown in Fig 4d) and Extended Data Fig. 6 were taken when the virtual barrier voltage was lower, and we

estimate the tunnel coupling between dots 1 and 2 in those case to be about 18 ueV. The tunnel coupling between dots 3 and 4 was not measured. We have added these remarks to the figure captions.

5. The measurement fidelity in this manuscript and the Ge hopping qubit papers is low compared to state of the art. Are there fundamental challenges to achieving higher measurement fidelity for this qubit encoding with tilted quantization axes and magnetic field gradients that the field needs to know about and solve or were the readout results here purely limited by technical challenges?

We believe both factors are at play here. At the high-field operating condition, we predict the quantization axis tip between the PSB dots is a few degrees, yet we were still only able to achieve visibilities of about 60-70% at best. We suspect the gate geometry and electrostatic tunability of the 2x2 may be the limiting factor, as our experience is that higher visibility is consistently easier to achieve in linear silicon arrays with the same manual tuning heuristics.

Due to the low visibility with limited quantization axis tips, we did not rigorously characterize changes in the readout behaviour when moving to low-field operation with larger quantization axis tips. However, the challenges arising in germanium should be similar here, where the energy level diagram will be sensitive to the Zeeman energy difference and the quantization axis tip. We did not measure the quantization axis tip between the dots used for PSB, though our simulations suggest it could be around 20 degrees (though this estimate is quite sensitive to dot location). Such a configuration will give rise to an S-T- avoided crossing that will play a role in the initialization and readout of spins with Pauli spin blockade. In the future, this could be taken advantage of with the appropriate magnetic field engineering to initialize and read the $|\text{down,down}\rangle$ spin state as is often done for hole spin qubits. Finally, we remark that future spin qubit designs could make use of separate zones for readout and control, which are individually optimized by e.g. local nanomagnets.

We have expanded the discussion on PSB readout limitations in the Methods section on “Initialization, Control, and Readout” to provide a more complete discussion of these points.

6. How long does it take for the B43, P3, B23 induced crosstalk on the sensor to settle out? Why do baseband pulses on P3 during the control sequence impact the readout portion which is several microseconds later? Would the same thing be observed on a charge

sensor wired for low frequency conductance measurements instead of RF reflectometry? Does this have implications for scaling this readout approach?

The timescale of this effect was not characterized in detail, and our empirical observations give a wide bound: though the effect persists through the readout integration, subsequently uploaded measurements do not appear to be affected. The timescale therefore falls between a few tens of microseconds and less than a few seconds. We don't have a detailed understanding of the effect beyond the correlation that the gates with the most cross-capacitance in the fanout (as starred in Extended Data Figure 1) seem to be the worst offenders, and the circuitous routing was designed to accommodate the fast line distribution of the PCB used to measure the sample. Our suspicion is that their pulses capacitively couple to an accumulation gate which strongly affects the adjacent sensor, and this would be the case whether using RF reflectometry or DC current. The effect seems particularly pronounced in this device, and we therefore suspect that these problems can be avoided in future designs. We have added this statement in line 906.

7. Is the reduction in visibility for the rabi oscillations in Extended Data Fig. 2 due to a similar effect? Why is the visibility reduced with a bias towards odd parity measurement outcomes?

After checking our measurement log, we confirmed that the initialization and readout protocol was recalibrated between taking the data of panel a) and panels b-d), which have similar visibilities. We have added this clarification in the caption of Extended Data Figure 2. We did not study this systematically to know if the higher driving amplitude correlated with the lower visibilities, or if the initialization/readout pulse was set to a suboptimal point in the window for the higher-power experiment. Regarding the consistent bias towards odd parity measurement outcomes, this is also an unresolved feature of PSB in this particular device that we observed both at high and low magnetic field operation.

Reviewer #2 (Remarks to the Author):

Key results:

In this manuscript, Unseld et al. demonstrated that micromagnet-based EDSR control of single-electron spins, traditionally used in 1D arrays of gate-defined quantum dots (QDs), is also feasible in a small 2D QD array in silicon. They confirmed this in a 2x2-QD array, additionally showcasing nearest-neighbor exchange control. They benchmarked the performance of the four-qubit system, particularly in relation to off-resonant drive-induced

crosstalk. Furthermore, by utilizing the same system as a two-qubit setup, they demonstrated qubit operation via hopping gates enabled by engineered magnetic stray fields. They also analyzed the qubit coherence, fidelity, and crosstalk associated with the hopping gate. They obtained high gate fidelities with both the approaches: a lower bound of 99.50% for the hopping gate and an average one of 99.54% for the resonant gate. Finally, they proposed a nanomagnet pattern to enable hopping control across an arbitrarily large 2D array.

Validity:

The validity and robustness of the data interpretation show no flaws, and the derived conclusions are sound.

Significance:

The exploitation of EDSR in a 2x2 Si QD array of single-electron spins, as reported in this manuscript, represents a pioneering yet crucial step toward the application of EDSR in scaled-up systems. Despite the challenges posed by microwaves—particularly heating effects and PIRS—which are likely to remain limiting factors, the introduction of hopping gates offers a promising alternative. Hopping gates eliminate the need for microwaves, thereby mitigating heating effects and enhancing scalability. The engineered micromagnets, with their specific design presented in this work, show significant potential in enabling hopping gates in larger 2D arrays, opening new possibilities for future advancements in quantum computing in silicon.

Data and methodology:

The validity of the approach is strong, and the quality of the presentation is excellent. Moreover, the supplementary notes are very helpful in providing explanations for the experimental results from a theoretical/modeling perspective. The quality of the results is very high, with only a minor weakness in qubit visibility for the hopping gates shown in Figure 1c. However, the authors have already provided a satisfactory explanation for the degraded readout signal in Section V of the manuscript.

Analytical approach:

The analytical approach is consistently strong throughout the manuscript, both in the resonant gate and hopping gate sections.

We thank the reviewer for their very kind words on the quality and significance of our manuscript. We agree that the readout quality is one of the main weaknesses of this work, and some further discussion on this topic has been added in response to points raised by

Reviewer #1.

Suggested improvements:

Here are some minor comments on the Extended Data Figures:

I would include the colorbar for the experimental (and fitted) qubit visibility in Figures 3 and 4. Moreover, I recommend adding the corresponding colorbar(s) to the graphs including the simulated magnetic fields shown in Figures 9(a), (c), (e), and (g).

We have added the suggested color bars to Extended Data Figures 3, 4, and 9.

Clarity and context:

The text is extremely clear and accessible, and the results are provided with great context and consideration of previous work.

References:

The manuscript appropriately references both previous and state-of-the-art literature.